# From Plaques to Pathways in Alzheimer’s Disease: The Mitochondrial-Neurovascular-Metabolic Hypothesis

**DOI:** 10.3390/ijms252111720

**Published:** 2024-10-31

**Authors:** Sarah Kazemeini, Ahmed Nadeem-Tariq, Ryan Shih, John Rafanan, Nabih Ghani, Thomas A. Vida

**Affiliations:** Kirk Kerkorian School of Medicine at UNLV, 625 Shadow Lane, Las Vegas, NV 89106, USA; kazems1@unlv.nevada.edu (S.K.); nadeemta@unlv.nevada.edu (A.N.-T.); shihr2@unlv.nevada.edu (R.S.); rafanj1@unlv.nevada.edu (J.R.); ghani@unlv.nevada.edu (N.G.)

**Keywords:** Alzheimer’s disease, amyloid-beta peptide, mitochondrial dysfunction, neurovascular regulation, metabolic disturbances, blood–brain barrier, neurodegeneration, oxidative stress

## Abstract

Alzheimer’s disease (AD) presents a public health challenge due to its progressive neurodegeneration, cognitive decline, and memory loss. The amyloid cascade hypothesis, which postulates that the accumulation of amyloid-beta (Aβ) peptides initiates a cascade leading to AD, has dominated research and therapeutic strategies. The failure of recent Aβ-targeted therapies to yield conclusive benefits necessitates further exploration of AD pathology. This review proposes the Mitochondrial–Neurovascular–Metabolic (MNM) hypothesis, which integrates mitochondrial dysfunction, impaired neurovascular regulation, and systemic metabolic disturbances as interrelated contributors to AD pathogenesis. Mitochondrial dysfunction, a hallmark of AD, leads to oxidative stress and bioenergetic failure. Concurrently, the breakdown of the blood–brain barrier (BBB) and impaired cerebral blood flow, which characterize neurovascular dysregulation, accelerate neurodegeneration. Metabolic disturbances such as glucose hypometabolism and insulin resistance further impair neuronal function and survival. This hypothesis highlights the interconnectedness of these pathways and suggests that therapeutic strategies targeting mitochondrial health, neurovascular integrity, and metabolic regulation may offer more effective interventions. The MNM hypothesis addresses these multifaceted aspects of AD, providing a comprehensive framework for understanding disease progression and developing novel therapeutic approaches. This approach paves the way for developing innovative therapeutic strategies that could significantly improve outcomes for millions affected worldwide.

## 1. Introduction

Alzheimer’s disease (AD) is a progressive neurodegenerative disorder that has received significant attention in modern medicine. In the United States alone, 6.7 million people aged 65 and older are currently living with AD, and this number may reach 13.8 million by 2060 [1]. The global prevalence of AD and dementia is rapidly increasing, with estimates ranging from 26.6 million in 2006 to 51.6 million in 2019 [2,3]. Projections suggest this number could quadruple by 2050, reaching 115–135 million cases worldwide [4,5]. The increasing prevalence of AD presents a growing public health challenge, with projected costs for long-term health care reaching over one trillion dollars in 2050 [6].

Age is the most significant risk factor, with prevalence increasing from 3% to 32% between ages 65 and 85 [7]. Life expectancy differs across sex and race, which may influence AD distribution in different populations. In the United States, women have a higher life expectancy (81 years) compared to men (76 years), which partially explains the higher prevalence of AD in women [8]. Additionally, racial and ethnic differences in life expectancy may impact AD prevalence. For example, life expectancy for Black individuals is lower (74.8 years) on average compared to white individuals (78.8 years), but Black women, who are at a higher risk for AD due to vascular factors and social determinants of health, have a life expectancy of 78.1 years [9,10]. This highlights the complex interaction between life expectancy, sex, race, and AD prevalence.

AD disproportionately affects women, with nearly two-thirds of AD patients being female [8,11]. Sex differences in AD risk factors include brain structure, stress responses, hormones, genetics, inflammation, and vascular disorders [12,13]. The prevalence of AD varies among racial and ethnic groups, and women, particularly Black women, face higher AD risk due to vascular factors and social determinants of health [9,10]. The understanding of AD has significantly evolved since Alois Alzheimer first described the disease in 1906 after identifying neurofibrillary tangles and plaques in the brain of a 50-year-old female patient [14]. His discovery sparked extensive research into these neuropathological findings, particularly amyloid plaques and neurofibrillary tangles [15].

AD manifests as a progressive neurodegenerative disorder, leading to cognitive decline, memory loss, and impaired daily functioning [16,17,18]. The disease’s pathology involves amyloid-beta (Aβ) peptide deposition, tau hyperphosphorylation, and mitochondrial dysfunction, collectively contributing to its progression [19]. In addition to Aβ and tau hyperphosphorylation, other fibrils, including those formed by the S100A family and alpha-synuclein, contribute to the pathology of AD. These fibrils are influenced by the ionic concentrations and redox conditions within brain tissues. S100A9 is a member of the S100A family, and it forms amyloid-like fibrils under these conditions, contributing to neuroinflammation [20]. Furthermore, alpha-synuclein forms amyloid fibrils under altered ionic strength and protein concentration, although it is primarily associated with Parkinson’s disease [21]. These fibril formations, in addition to Aβ and tau, demonstrate the multifactorial nature of AD pathogenesis.

AD disrupts hippocampal neurogenesis, essential for learning and memory [22], and alters critical neurotransmitter systems, including cholinergic, adrenergic, and glutamatergic networks [23]. Clinicians rely on clinical parameters, biomarkers, and imaging methods for diagnosis, with neuroimaging playing a crucial role in early detection [24,25]. Aβ plaques and tau tangles mark the disease’s heterogeneous clinical presentations and vary between genetic and sporadic forms, often involving aging hallmarks like genomic instability and epigenetic dysregulation [26,27,28,29]. Despite extensive efforts to target Aβ and tau, most therapeutic trials have failed [30,31,32], underscoring the need for new approaches that may involve a multifactorial strategy addressing diet, lifestyle, and other factors to prevent or reverse cognitive decline more effectively [24,30,33].

Recent research highlights the significant role of diet, ethnicity, and social determinants in AD development. Dietary patterns, such as the Mediterranean, DASH, and MIND diets, have demonstrated protective effects against cognitive decline. Research led by Martha Clare Morris has shown that adherence to the MIND diet, rich in antioxidants, omega-3 fatty acids, and neuroprotective nutrients, is associated with slower cognitive decline [34,35,36]. These diets can improve brain health, while diets high in saturated fats and processed foods have been linked to increased AD risk [37].

Cultural and ethnic differences further modulate AD risk. Black and Hispanic populations show a higher prevalence of AD compared to white individuals, which may be attributed to genetic predispositions, differences in health behaviors, and disparities in healthcare access [38]. Epidemiological studies highlight these disparities, underscoring how socioeconomic status, educational attainment, and access to healthcare contribute to elevated AD risk among minority groups, revealing critical insights into how social and structural factors intersect with genetic and lifestyle elements in AD prevalence [39]. Socioeconomic status can limit access to healthier dietary options, emphasizing the need for targeted interventions that account for these social determinants [40]. Social determinants, including education, income, and healthcare access, are intricately linked to both the onset and progression of AD, as individuals with lower education levels or limited healthcare access may experience delayed diagnosis and poorer disease management [37,40]. Accurate and timely diagnosis remains essential for optimizing patient care and developing potential disease-modifying therapies [41].

Traditionally, the amyloid cascade hypothesis, which posits that Aβ accumulation initiates AD pathogenesis, has been a central focus of AD research and drug development. However, recent clinical trials targeting Aβ have shown limited efficacy, prompting a re-evaluation of this hypothesis [42,43]. While some anti-Aβ antibodies can reduce Aβ plaques, their clinical benefits remain questionable [44]. This has led to exploring alternative models in AD pathogenesis, recognizing the disease’s multifactorial nature [45,46] and suggesting that the disease likely arises from the interplay of several interrelated pathways.

These developments have led us to propose the Mitochondrial–Neurovascular–Metabolic (MNM) hypothesis, which integrates mitochondrial dysfunction, impaired neurovascular regulation, and systemic metabolic disturbances into a cohesive framework that may offer a more comprehensive understanding of AD pathology and open new avenues for therapeutic intervention. This review critically examines the current evidence supporting the MNM hypothesis and its implications for the future of AD research and treatment. We will explore the biochemical and genetic foundations of the amyloid cascade hypothesis, the challenges that have prompted its reconsideration, and the integration of mitochondrial, neurovascular, and metabolic perspectives in understanding AD. We aim to present a unified approach that could pave the way for more effective interventions in combating this devastating disease.

## 2. Amyloid Cascade Hypothesis: Revisiting a Traditional Framework

### 2.1. Biochemical Mechanisms and Pathways

The amyloid cascade hypothesis suggests that the accumulation of Aβ in the brain is the central event that triggers the pathological cascade leading to Alzheimer’s disease [47]. This cascade involves the formation of Aβ plaques, which initiate a series of events, including neurofibrillary tangles, inflammation, synaptic dysfunction, and, ultimately, neuronal death. Aβ accumulation triggers a cascade of events, including tau hyperphosphorylation, neuroinflammation, and synaptic dysfunction [48,49]. Evidence for this view is substantial. Aβ derives from the amyloid precursor protein (APP) cleavage through the amyloidogenic pathway, where enzymes such as β-secretase and γ-secretase sequentially cleave the protein [50,51,52]. This process forms Aβ of various lengths, with Aβ42 particularly prone to aggregation and plaque formation [53]. In addition, the supersaturation concept suggests that balance disruptions between production and clearance of Aβ influence its aggregation, leading to accumulation and deposition [54]. In particular, the effects of Aβ on synapse formation and neurite outgrowth are concentration-dependent [55], and it can decrease the expression of memory-related receptors such as N-Methyl-D-Aspartate receptors (NMDARs) [56]. Furthermore, accumulating Aβ has other notable implications, as it can decrease dendritic spine density and subdue spine motility [57]. This perspective requires a comprehensive understanding of Aβ dynamics and its interaction with other cellular processes [54]. Widely accepted biochemical pathways and mechanisms reveal complex relationships, including Aβ accumulation, neuroinflammation, tau hyperphosphorylation, and dysregulated calcium homeostasis (Figure 1).

### 2.2. Limitations and Emerging Challenges

The Aβ hypothesis has dominated the focus of AD research for decades, driving therapeutic interventions. Despite extensive efforts to develop anti-Aβ therapies, these treatments have largely failed to demonstrate efficacy in slowing cognitive decline, with numerous clinical trials yielding inconclusive results [58,59]. Many trials fail to produce conclusive benefits, calling into question the relevance of Aβ as the central therapeutic target [60,61]. Even recent approvals of antibodies such as Aducanumab and Lecanemab show limited effectiveness in reducing Aβ load without translating into significant cognitive improvements [43]. This failure has led to growing skepticism about the hypothesis, prompting the exploration of alternative models and approaches [62,63]. These failures reflect insufficient target engagement, disease heterogeneity, and the complex, nonlinear relationship between Aβ deposition and cognitive decline [64].

The Aβ burden does not always correlate with cognitive decline. Some individuals can exhibit high levels of Aβ without cognitive impairment, while others continue to decline cognitively even after plaque removal [65]. A complex relationship occurs between Aβ burden and cognitive decline in older adults. While Aβ positivity predicts cognitive decline [66,67], some individuals with high Aβ levels maintain normal cognition [68,69]. Factors associated with cognitive resilience despite Aβ burden include higher cerebrospinal fluid Aβ42 levels [69] and enrichment of specific synaptic proteins, but some individuals experience cognitive decline with minimal Aβ pathology [70]. This has suggested that Aβ plaques might result from neurodegeneration rather than a causative factor [71]. Early cognitive changes may precede detectable Aβ accumulation [72], and plasma Aβ42/40 ratios might be a biomarker for identifying those at risk of decline [73]. These findings underscore the importance of considering multiple factors beyond Aβ burden when assessing cognitive risk and developing interventions for Alzheimer’s disease [74]. Moreover, the complexity of AD pathogenesis has become more apparent, with emerging studies highlighting the roles of innate immunity, inflammation, and environmental factors [75,76].

This evolving understanding has given rise to a more comprehensive, probabilistic model that integrates genetic and environmental factors alongside Aβ pathology, shifting the focus from a single pathway to a multifactorial perspective [44,61,62]. For example, traumatic brain injuries (TBIs) can be linked to AD pathology, as studies in mice show that mild traumatic brain injuries (mTBIs) can cause blood–brain barrier (BBB) leakage, Aβ pathology, and subsequent cognitive impairment [77]. In humans, abnormal deposits of neurofibrillary tangles and Aβ plaques from repetitive mTBIs, especially in military personnel, are associated with chronic traumatic encephalopathy [78]. These findings suggest that trauma-related neurodegenerative changes may share pathological processes with AD, expanding the scope of Aβ pathology beyond Aβ accumulation alone. The complex interplay of multiple pathways requires a broader approach to understanding and addressing the disease [61].

## 3. The Mitochondrial–Neurovascular–Metabolic (MNM) Hypothesis: An Integrated Perspective Focusing Forward

Amid the pleiotropy of AD and the Aβ cascade framework, we propose a novel premise, the Mitochondrial–Neurovascular–Metabolic (MNM) hypothesis. The MNM hypothesis offers a critical departure from the amyloid-centric view. This notion integrates three critical but often separately studied pathways: (1) mitochondrial dysfunction, (2) neurovascular dysregulation, and metabolic disturbances of glucose in the brain, elements often studied in isolation. Unlike the amyloid hypothesis, which fails to account for the heterogeneous manifestations of AD, the MNM hypothesis provides a framework that acknowledges the interplay between cellular energy deficits, vascular integrity, and systemic metabolic health. This perspective explains the diverse clinical presentations of AD. Recently, views combining mitochondrial dysfunction, neurovascular breakdown, and metabolic disturbances have emerged in understanding AD [79]. However, most hypotheses still link these factors to the amyloidogenic model, treating them as secondary contributors alongside Aβ and tau pathologies. In contrast, the MNM hypothesis positions mitochondrial, neurovascular, and metabolic dysfunctions as AD’s primary, interdependent drivers. This approach offers a more unified model that surpasses the limitations of an amyloid-centric perspective.

The Mitochondrial–Neurovascular–Metabolic (MNM) hypothesis offers a comprehensive model for Alzheimer’s disease (AD) pathogenesis, emphasizing the interconnectedness of mitochondrial dysfunction, neurovascular unit breakdown, and metabolic disturbances [80].

### 3.1. Mitochondrial Dysfunction

Mitochondrial dysfunction reduces ATP production, increases reactive oxygen species (ROS), disrupts calcium homeostasis, and contributes to neuronal damage [81,82,83,84]. The increased oxidative stress and neuronal damage impair synaptic function, contributing to cognitive decline [85]. Mitochondrial dynamics, including fission, fusion, and mitophagy, are dysregulated in AD, affecting energy production and quality control [86,87]. The brain’s high energy demand makes it particularly vulnerable to mitochondrial impairment [88]. Recent research has also highlighted the role of mitochondrial dysfunction in microglial activation and neuroinflammation in AD [89]. Understanding these mechanisms is crucial for developing novel therapeutic strategies targeting mitochondrial function to treat AD and other neurodegenerative diseases [90].

### 3.2. Neurovascular Dysregulation

Neurovascular dysfunction, including blood–brain barrier deterioration and cerebral hypoperfusion, precedes Aβ accumulation and exacerbates AD progression [91,92]. The breakdown of the neurovascular unit (NVU) compromises cerebral homeostasis, allowing neurotoxic substances to infiltrate the brain and accelerate AD progression, establishing a direct link between vascular health and neurodegeneration [93]. The NVU, comprising endothelial cells, astrocytes, pericytes, and neurons, maintains cerebral homeostasis and blood–brain barrier (BBB) integrity [94,95]. In AD, NVU dysfunction leads to BBB breakdown, allowing neurotoxic substances to infiltrate the brain, accelerating neuronal injury, and inducing neuroinflammation [96,97]. This creates a detrimental feedback loop, compromising the BBB [98]. Impaired cerebral blood flow reduces oxygen and glucose delivery, exacerbating neuronal energy deficits and promoting mitochondrial dysfunction [79,99]. Microglia-mediated NVU dysfunction contributes to AD progression through various mechanisms, including promoting neuroinflammation and oxidative stress [94]. The complex interactions between Aβ, tau, and NVU components highlight the multifaceted nature of AD pathogenesis, emphasizing the need for novel therapeutic approaches targeting NVU dysfunction [97,100].

### 3.3. Metabolic Impairment

Glucose hypometabolism and insulin resistance in the brain are increasingly recognized as a metabolic disorder that characterizes AD [101,102,103,104]. These metabolic disturbances precede clinical symptoms and contribute to neuronal energy deficits, mitochondrial dysfunction, and impaired Aβ clearance [85,105]. Glucose uptake and utilization are reduced due to the downregulation of glucose transporters like GLUT1 and GLUT3 [105,106]. The resulting energy deficiency affects synaptic function and neuronal survival [107]. Importantly, type 2 diabetes mellitus shares common pathophysiological mechanisms with AD, leading to the view of AD as “type 3 diabetes” [108]. These metabolic alterations offer potential targets for early diagnosis and therapeutic interventions, including strategies to improve insulin signaling in the brain [109].

Mitochondrial dysfunction, neurovascular dysregulation, and metabolic impairment in the brain accelerate neurodegeneration and cognitive decline, ultimately leading to the accumulation of Aβ. These interconnected pathways trigger a cascade of events that culminate in severe neuronal damage, demonstrating the need for a model that accurately captures the multifactorial nature of Alzheimer’s disease (AD). The Mitochondrial–Neurovascular–Metabolic (MNM) hypothesis fills this role as a transformative model, offering an integrative perspective that surpasses the limitations of the amyloid-centric theory. This hypothesis reveals the complex interplay of disturbances, providing a more comprehensive understanding of AD pathology and opening the door for therapeutic strategies that target multiple pathways concurrently. The MNM hypothesis sets a clear foundation for future AD research and interventions, offering a promising direction for more effective treatments. A detailed comparison between the Aβ and MNM hypotheses is presented in Table 1.

## 4. Mitochondrial Dysfunction in Alzheimer’s Disease: Mechanisms and Implications

### 4.1. Mitochondria: Essential Roles in Neuronal Health

An essential component of the MNM hypothesis is mitochondrial dysfunction. Mitochondria are the primary energy producers in neurons and have various roles in neuronal health. These organelles convert glucose and oxygen into ATP for cellular activities, essential for multiple neuronal functions such as neurotransmitter release and membrane potential maintenance [110,111]. The rate of neuronal cell respiration is notably higher than in other cells, emphasizing the critical role of mitochondria in neuronal energy metabolism [112]. In AD, impaired mitochondrial function leads to disrupted ATP production, increased oxidative stress, and impaired calcium homeostasis, contributing to neuronal damage and synaptic dysfunction [113]. Neurotransmitter release, generation of nerve impulses, and regulation of membrane potentials are crucial for proper communication between neurons and the overall functioning of the nervous system. These disturbances exacerbate the pathophysiological processes underlying AD, reinforcing the idea that mitochondrial dysfunction is not merely a consequence but a driver of disease progression, as integrated within the MNM hypothesis.

Considering the integral role of mitochondria in neuronal health, AD patients have notably altered mitochondrial function compared to healthy individuals. 3xTg-AD mice show significant hypometabolic changes in brain metabolism, supporting the idea that impaired oxidative phosphorylation leads to neurodegeneration [114]. Increased metabolic activity among those with AD is also attributed to increased mitochondria-associated endoplasmic reticulum membranes due to increased levels of C99, the C-terminal processing product of the amyloid precursor protein [115]. These reports have prompted the investigation of the organelle’s role in understanding AD development and progression.

### 4.2. Mitochondrial Cascade Hypothesis: A Pathogenic Model

Because this organelle plays a critical role in fulfilling the high metabolic demands of neurons and maintaining neuronal health, the mitochondrial cascade hypothesis is needed to understand AD pathogenesis. According to the hypothesis, mitochondrial dysfunction is a critical step in AD progression, facilitating an understanding of the defects seen in mitochondrial components in AD. Primary mitochondrial alterations trigger cellular processes, including increased reactive oxygen species (ROS) production, oxidative stress, and impaired energy metabolism. These events are thought to promote Aβ accumulation and induce tau hyperphosphorylation, disrupting synaptic function and eventual neuronal death. Extensive research has been conducted to explore the possibility that mitochondria may contribute to or drive the clinical presentation of AD.

### 4.3. Altered Mitochondrial Fission and Fusion Dynamics in Alzheimer’s Disease

Abnormal mitochondrial dynamics are significant factors in AD [116]. Mitochondria regularly undergo fission and fusion, essential for maintaining mitochondrial functions and inheritance [117]. Fission is necessary for cytokinesis [118], while mitochondrial fusion facilitates content mixing within a mitochondrial population [119]. If the rates of fusion and fission are unbalanced, mitochondrial morphology and physiology are affected in neurons from AD-biopsied brains and cells with mutant AβPP; excessive fission results in round, structurally compromised mitochondria [120]. In Aβ-treated neurons and AD-patient-derived fibroblasts, increased interaction between mitochondrial fission proteins promotes excessive mitochondrial fission. Inhibition of this interaction with peptide P110 improves mitochondrial function and cognitive performance in AD models [121].

A phenomenon termed mitochondrial fission arrest has also been observed in AD, where altered brain energetics induce the formation of elongated mitochondria called “mitochondria-on-a-string” (MOAS). This arrest, resulting from increased recruitment of Drp1, a protein involved in mitochondrial fission, and reduced GTPase activity, may be a compensatory response to bioenergetic stress [122]. These findings suggest that both excessive fission and fission arrest contribute to the structural damage seen in AD, supporting the hypothesis that an imbalance of fusion and fission is critical in the disease’s development.

### 4.4. Mitophagy Impairment and Mitochondrial Dysfunction in Alzheimer’s Disease

Mitophagy, a specific form of autophagy that removes damaged mitochondria, is significantly impaired in AD [123]. It eliminates defective organelles to maintain cellular health, especially mitochondria where reactive oxygen species (ROS) have been damaged beyond repair. Levels of mitophagy are reduced by up to 50% in post-mortem hippocampal brain samples in AD patients compared to age-matched cognitively healthy ones [124]. AD-affected brains have also been identified to have a downregulation of the expression of various proteins known to participate in autophagy and mitophagy processes, including Optineurin (OPTN), ATG5, PI3K class III, ULK1, AMBRA1, BNIP3, BNIP3L, VDAC1, and VCP/P97 [125]. A reduction in activity among mitochondrial complexes I, II, and V have been identified in the entorhinal cortex (EC), suggesting that mitophagy failure and mitochondrial dysfunction occur primarily within vulnerable brain regions [126,127]. Mitochondrial damage is not the sole characteristic of AD. However, its harmful effects increase as dysfunctional mitochondria accumulate because normal cellular processes fail to eliminate ineffective components.

### 4.5. Oxidative Stress and Reactive Oxygen Species Production

Mitochondrial dynamics significantly impact ROS production and bioenergetic failure. Structurally and functionally impaired mitochondria lead to reduced ATP production and increased ROS generation. Key oxidative phosphorylation enzymes, such as cytochrome oxidase, show reduced activity in AD, correlating with clinical severity and plaque counts [128]. Proximity to the electron transport chain makes mitochondrial DNA susceptible to oxidative damage, exacerbating ROS production and bioenergetic failure [129]. AD neurons exhibit increased mitochondrial fragmentation, reduced levels of fusion proteins (Mfn1 and Mfn2), and increased levels of fission proteins, which enhances oxidative stress and neuronal damage [130]. These impairments suggest a plausible correlation between impaired mitochondrial structure and neurological deficits, which characterize the clinical presentation of AD.

While mitochondria are a significant source of ROS, they also play a crucial role in ROS detoxification. The brain’s high oxygen consumption makes it particularly vulnerable to ROS-induced damage [131]. An estimated 1–2% of oxygen is converted to ROS, and most free radicals are produced in mitochondrial respiration, especially during electron flow in complexes I, II, and III of the electron transport chain [132]. Therefore, the balance between ROS production and elimination is essential to prevent oxidative stress, which can lead to neuronal damage and is implicated in various neurodegenerative diseases [133]. In AD, oxidative damage to neuronal macromolecules, such as lipid peroxidation products (e.g., 4-hydroxynonenal, malondialdehyde), protein oxidation markers (e.g., protein carbonyls, 3-nitrotyrosine), and DNA/RNA oxidation markers, is elevated across various regions [134]. Antioxidants, responsible for neutralizing ROS, are also observed at decreased levels among AD patients; these include uric acid, superoxide dismutase, and vitamins C and E [135]. Consequently, the imbalance between ROS production and elimination is evident in neurological disorders such as AD. Antioxidant therapies targeting mitochondria, such as acetyl-L-carnitine and R-alpha-lipoic acid, show promise in preclinical studies [136]. However, clinical trials with antioxidants have yielded inconsistent results, highlighting the need for more integrated approaches [137]. The role of mitochondria in the MNM hypothesis and AD pathogenesis is depicted in Figure 2.

## 5. Neurovascular Dysregulation in Alzheimer’s Disease

### 5.1. Neurovascular Health: The Foundation of Brain Function

In addition to mitochondrial dysfunction, the neurovascular unit (NVU) is essential in maintaining cerebral homeostasis. Impairments in neurovascular regulation are recognized as contributing factors to the pathophysiology of AD. Therefore, it is beneficial to understand the association of vascular health with neurodegeneration. Cerebral vascular health and systemic metabolism are often considered in isolation, although these aspects are related and typically altered in AD patients [138,139].

Regarding cerebral vascular health, the neurovascular unit (NVU) contributes to brain homeostasis and comprises neurons, astrocytes, endothelial cells, and the extracellular matrix [91]. The BBB, an integral part of the NVU, utilizes tight junction proteins, including claudin-5 and occludin, to prevent the entry of pathogens and other blood-related products into the brain [140,141]. Vascular alterations in brain samples from AD patients show neurotoxic substances, such as fibrinogen, in the parenchyma due to their ability to pass the BBB [142,143].

The NVU regulates cerebral blood flow (CBF) through neurovascular coupling (NVC). This process ensures active brain regions receive adequate blood flow for oxygen and nutrients [144]. During heightened neuronal activity, the NVU meets the brain’s increased energy demands among healthy individuals [145]. Early alterations in cerebral vascular health, including reduced CBF and BBB disruption, may occur before the appearance of Aβ plaques [146].

### 5.2. Astrocyte-Mediated NVU Maintenance and Its Breakdown in Alzheimer’s Disease

Astrocytes provide metabolic substrates to neurons, maintain ion balance, and play a role in Aβ plaque formation, thereby maintaining the NVU. Under cellular stress, astrocytes produce Aβ, and its presence can induce the release of pro-inflammatory cytokines from these cells [147]. Aggregating large amounts of Aβ proteins causes Aβ-mediated stress, leading to mitochondrial swelling and excessive astrocyte fission [148]. In response to this stress, astrocytes shift their energy metabolism to glycolysis and peroxisomal-based fatty acid β-oxidation [148]. Additionally, astrocytes respond to neurotransmitters, increasing intracellular calcium levels and releasing gliotransmitters that activate neuronal receptors [149]. Astrocyte–neuron signaling remains preserved during healthy aging but becomes dysregulated in *APP/PS1* mice, a genetically modified model that overexpresses human amyloid precursor protein (APP) and presenilin-1 (*PS1*) genes. This dysregulation contributes to Aβ plaque formation and disruptions in energy metabolism [149]. Understanding astrocyte–neuron interactions can enhance our knowledge of how dysfunctions in the NVU are linked to pathological mechanisms in AD, including the formation of additional Aβ plaques and metabolic shifts.

### 5.3. Cytokine-Mediated Blood–Brain Barrier Dysfunction in Alzheimer’s Disease: Implications for Neurovascular and Metabolic Interactions

The integrity of the blood–brain barrier (BBB) is critically impaired in AD, affecting neuronal connectivity and synaptic functioning [150]. The BBB’s integrity in AD becomes significantly compromised, especially in regions like the cortex and hippocampus, leading to increased permeability [151]. The BBB also selectively removes metabolic waste products from the brain [152]. The altered expression of tight junction proteins, including occludin and claudin-5, increases the permeability of the BBB [153,154]. Due to this increased permeability, neurotoxic molecules can enter the brain, leading to neuronal damage and cognitive decline [155]. Maintaining BBB integrity is, therefore, essential for regulating interstitial fluid composition and selectively removing wastes, highlighting the fundamental role of vascular permeability in the development and symptomatology of AD. The MNM hypothesis uniquely incorporates this dysfunction, focusing on how BBB impairment contributes to a vicious cycle of neurodegeneration, oxidative stress, and metabolic disruption. In contrast to the amyloid hypothesis, which overlooks this critical aspect, the MNM model emphasizes that addressing BBB integrity is not merely a secondary factor but a central element in understanding AD’s progression. Thus, the hypothesis suggests that therapeutic strategies targeting BBB restoration could effectively intervene in AD’s pathogenesis, enhancing the overall therapeutic potential compared to more amyloid-centric approaches.

Cytokines are pivotal in Alzheimer’s disease (AD) pathogenesis, particularly in modulating blood–brain barrier (BBB) function and neuroinflammation [156]. However, while a clear link exists between cytokine activity and AD progression, a critical examination suggests that their role is far more complex and interwoven with other aspects of the disease, according to the MNM hypothesis.

Pro-inflammatory cytokines such as IL-1β, TNF-α, and IL-6 increase BBB permeability, disrupt tight junctions, and reduce the expression of Aβ efflux transporters, potentially contributing to Aβ accumulation [156,157]. These findings suggest cytokine-driven BBB disruption is a significant factor in AD pathogenesis, consistent with the MNM hypothesis’s emphasis on neurovascular dysregulation. Additionally, these cytokines activate microglia, leading to further inflammation and neurodegeneration [158]. However, the degree to which cytokines directly drive AD pathology versus serving as secondary agents in response to existing disease processes is still debated. This ambiguity necessitates a broader perspective, as cytokine activity may result from mitochondrial dysfunction or other systemic metabolic disturbances rather than an isolated pathogenic event.

Conversely, anti-inflammatory cytokines such as IL-4 and IL-10 are proposed to have neuroprotective roles by suppressing pro-inflammatory cytokine production, potentially mitigating AD progression [159,160]. While this suggests a therapeutic potential for modulating cytokine activity, it is crucial to recognize that cytokine signaling at the BBB involves complex interactions with lipid rafts and nuclear hormone receptors, which influence transporter function and lipid metabolism [161]. This intricate signaling underscores that targeting cytokine activity alone might not be sufficient; instead, it should be considered in conjunction with interventions that address mitochondrial health and metabolic regulation [162,163], emphasized in the MNM hypothesis.

Recent studies have further highlighted the multifaceted nature of AD pathogenesis, indicating that neuroinflammation and BBB dysfunction are interconnected with other factors, such as the gut microbiome, which may influence AD pathology through cytokine modulation [164]. Moreover, age-related chronic inflammation and myelin dysfunction have been identified as upstream risk factors for AD, suggesting that cytokine activity is part of a broader inflammatory milieu contributing to disease progression [165,166]. These findings align with the MNM hypothesis’s recognition of systemic metabolic and vascular contributions to AD.

Additionally, while blood-based biomarkers, including cytokine profiles, are emerging as promising tools for earlier AD diagnosis [167], the direct translation of these findings into targeted therapies has yet to be fully realized. This gap reinforces the need for a more integrated therapeutic approach that addresses cytokine signaling in the context of mitochondrial dysfunction, neurovascular breakdown, and metabolic disturbances [168,169].

Activated microglia induce reactive astrocytes, which also play a role in neurotoxicity and BBB disruption [170]. This further highlights how cytokine-driven neuroinflammation contributes to the complex interplay between neurovascular dysfunction and mitochondrial impairment. Similarly, BBB breakdown is observed as an early sign of AD, suggesting that interventions aimed at preserving BBB integrity, possibly through modulating cytokine activity, could be a viable therapeutic strategy [171].

Understanding the interactions between cytokines, BBB function, and the more extensive network of mitochondrial and metabolic pathways is crucial for developing targeted therapies [172]. Cytokine modulation at the BBB is not a standalone phenomenon but rather part of a more extensive systemic process that involves the interplay of mitochondrial dysfunction, neurovascular regulation, and metabolic disturbances [161,163]. As such, targeting these interconnected pathways aligns more closely with the MNM hypothesis’s multifactorial approach to AD pathogenesis.

### 5.4. Neurovascular Coupling, Mitochondrial Dysfunction, and Endothelial Vulnerability

Disruptions in NVC affect the clearance of Aβ, worsening neuronal damage and advancing disease progression [173,174]. IGF-1 deficiency has also been identified to impair cerebral microvascular endothelial function, ultimately harming glutamate-mediated CBF responses and leading to the aging phenotype associated with cognitive impairment [175]. In an AD mouse model, NVC and hyperemia caused perivascular Aβ accumulation in the neocortex at earlier ages [176]. CBF and glucose utilization are also reduced in transgenic mice overexpressing the APP [177]. NVC is essential for practical brain function; alterations may be linked to neurological deficits. Metabolic and cerebrovascular abnormalities are potentially characteristic events in the pathogenesis of AD.

The impact of mitochondrial dysfunction on NVU components is significant, as endothelial cells are vulnerable to mitochondrial impairments. Endothelial cells rely on mitochondrial function to maintain BBB integrity. Hawkins et al. injected rats with streptozotocin to induce diabetes in their experiments [178]. They assessed BBB function by quantifying matrix metalloproteinase activity associated with the degradation of tight junction proteins [178]. It was concluded that losing tight junction proteins and hyperglycemia may increase BBB permeability. Furthermore, mitochondrial dysfunction within endothelial progenitor cells (EPCs) negatively affects the mature endothelial cells they differentiate into, impairing vascular function due to less effective endothelial repair [179]. Mitochondrial dysfunction within endothelial cells and their progenitors is crucial in compromising BBB integrity and overall vascular function, highlighting the importance of targeting mitochondrial health in preventing cerebrovascular contributions to AD. The role of the NVU in the MNM hypothesis and AD pathogenesis is depicted in Figure 2. A schematic representation of the existing biochemical routes that could lead to neurovascular dysregulation in AD is provided in Figure 3.

## 6. Comparative Analysis of Brain Metabolism: Healthy vs. Alzheimer’s-Affected

### 6.1. The Importance of Brain Metabolism in Cognitive Function

The relationship between mitochondrial and neurovascular dysfunctions significantly impacts overall brain metabolism. Brain metabolism, particularly glucose metabolism, is vital in maintaining cognitive function and neuronal health. In AD, studies using positron emission tomography (PET) with ^18^F-fluorodeoxyglucose (FDG) consistently show reduced cerebral metabolic rates of glucose (CMRglc) in critical brain regions, such as the hippocampus and posterior cingulate cortex, even in the disease’s early stages [106]. This reduction in glucose metabolism correlates with the severity of cognitive decline observed in AD patients [180]. Furthermore, cerebral glucose hypometabolism assessed with FDG-PET is now a biomarker for early AD detection. It can predict the conversion from mild cognitive impairment (MCI) to AD with reasonable accuracy [181]. AD profoundly impacts brain metabolism, differing significantly from the metabolic processes observed in a healthy brain. Unlike the amyloid hypothesis, which does not account for such metabolic disruptions, the MNM hypothesis integrates this phenomenon, emphasizing how glucose hypometabolism contributes to neuronal dysfunction and cognitive decline. This integration reinforces the role of energy deficits in AD pathogenesis and underscores the potential for targeted interventions that restore metabolic balance. Since cerebral glucose hypometabolism is a central feature, the MNM hypothesis offers a more comprehensive framework for understanding the disease’s multifaceted nature. This section reviews vital studies that compare these metabolic differences and explore the underlying mechanisms of metabolic dysfunction in AD.

### 6.2. Disrupted Glucose Metabolism in Alzheimer’s Disease

In a healthy brain, glucose metabolism is a primary energy source, with efficient glucose transport and utilization through glycolysis and oxidative phosphorylation. Glucose transporters like GLUT1 and GLUT3 are essential in ensuring a steady supply of glucose across the BBB and into neurons [182]. Moreover, astrocytes play a vital role in this process as they take glucose from the bloodstream, convert it to lactate, and shuttle it to neurons for energy use [107]. This astrocyte–neuron metabolic coupling is essential for maintaining energy homeostasis and supporting neuronal activity. In AD, glucose metabolism markedly declines, particularly in regions such as the frontal, temporal, and parietal cortices [183,184]. This hypometabolism is closely linked to disruptions in the brain’s energy supply processes, including the integrity of the NVU and astrocyte–neuron metabolic coupling [107]. The role of metabolic impairment in the MNM hypothesis and AD pathogenesis is depicted in Figure 2.

### 6.3. Insulin Resistance and Its Role in Alzheimer’s Disease Pathogenesis

Insulin signaling is crucial in healthy and AD brains [185]. Insulin is vital for glucose uptake in peripheral tissues, and its dysregulation in the brain affects neuronal function and survival. In healthy brains, insulin regulates various functions, including memory and synaptic plasticity [186], with the PI3K/Akt/GSK3β pathway being central to these processes, where Akt, a serine/threonine kinase, phosphorylates and inactivates GSK3β, promoting glycogen synthesis and supporting cellular health [187]. Insulin resistance in the brain reduces glucose uptake and utilization, exacerbating metabolic dysfunction and cognitive impairment [106]. As observed in AD models, impaired insulin signaling and insulin resistance lead to reduced phosphorylation in the PI3K/Akt/GSK3β pathway, contributing to cognitive decline and neurodegeneration [188,189]. Decreased activation of Akt results in the constitutive activity of GSK3β, which exacerbates metabolic dysfunction and further drives neurodegenerative processes. This insulin resistance–GSK-3β axis also affects mitochondrial function, energy metabolism, and cognitive performance [190,191], and recent evidence highlights the critical role of GSK3β in neuroinflammation and neurodegeneration [192].

Insulin resistance, commonly associated with type 2 diabetes, is also prevalent in AD, supporting a view of the disease as “type 3 diabetes” due to the shared features of impaired insulin signaling and glucose metabolism [193]. Galanin, a neuropeptide widely distributed in the CNS and PNS, plays a role in memory decline when overexpressed [194]. Galanin levels decrease after administration of two antidiabetic drugs, glibenclamide and pioglitazone, in a rat model [195]. These findings suggest that inhibiting galanin in the brain using antidiabetic drugs dramatically improves AD symptoms, supporting the hypothesis that insulin resistance plays a crucial role in AD pathogenesis [195].

The decline in cognitive abilities in AD is closely linked to the loss of synaptic function, especially within the hippocampus, a region with high insulin receptor (IR) expression and crucial for memory [186,196]. Impaired brain insulin signaling, or brain insulin resistance (BIR), is associated with cognitive decline and Alzheimer’s disease (AD) [197,198]. Neuroimaging studies reveal that insulin resistance affects brain regions vulnerable to AD pathology, including the temporal lobe and hippocampus [199]. Postmortem studies show altered insulin signaling in AD brains, with decreased vascular insulin receptors correlating with cognitive impairment and increased Aβ pathology [200,201]. In addition to GLUT1 and GLUT3 [105,106], the insulin-responsive glucose transporter GLUT4 may play a central role in hippocampal memory processes, and its reduced activation could underlie cognitive impairments in insulin resistance [202]. These findings suggest that targeting brain insulin signaling could be a potential therapeutic approach for AD.

This connection highlights the significance of insulin signaling in cognitive processes. Studies have demonstrated that insulin administration through intranasal and intracerebroventricular routes can enhance spatial memory, regulate hippocampal activity, and partially restore cognitive function in AD patients [203]. Disrupted insulin signaling can result in neuronal energy deficits, making neurons more susceptible to oxidative stress and damage (Figure 4). AD patients often exhibit reduced cerebral glucose metabolism, likely due to insulin signaling impairments and altered thiamine metabolism [204,205].

### 6.4. The Wnt Signaling Nexus: Unraveling Metabolic Dysfunctions in Alzheimer’s Disease

Wnt signaling pathways, known for their roles in embryogenesis and cancer, regulate brain glucose metabolism. Activation of Wnt signaling enhances glucose uptake and utilization in cortical neurons through increased pentose phosphate pathway activity [206,207]. Various pathways mediate this metabolic regulation, including Akt and nitric oxide signaling [206,208]. Wnt activation also facilitates glucose uptake and glycolysis in neurons, increasing cognitive function in AD mouse models [206,207]). This process involves the PI3K-AKT-mTOR pathway, which interacts with Wnt signaling to regulate cell cycle progression and metabolic reprogramming in cancer cells [209]. Wnt signaling, therefore, supports various aspects of brain function in a healthy brain, including neuronal growth and synaptic plasticity.

In AD, dysregulation of Wnt pathways impairs glucose uptake and utilization. Abnormal Wnt signaling affects critical metabolic protein expression and contributes to characteristic AD’s energy deficits. When considering these implications, therapeutic targeting of Wnt signaling may help restore metabolic function and slow disease progression [207]. Abnormal Wnt signaling, particularly in the canonical Wnt/β-catenin pathway, affects the expression of vital metabolic proteins and contributes to synaptic dysfunction. Reduced β-catenin levels in patients with presenilin-1 mutations and inhibition from AB and Dkk1 provide evidence for this dysfunction. Furthermore, Wnt signaling intersects with metabolic disorders such as obesity, lipoprotein metabolism, hypertension, and insulin resistance [204].

Building on the disrupted metabolic pathways in AD due to abnormal Wnt signaling and insulin resistance, the ketogenic diet offers a promising therapeutic avenue, addressing the brain’s impaired glucose metabolism. The ketogenic diet shifts the brain’s energy source from glucose to ketone bodies, which AD brains can metabolize more effectively. The shift enhances mitochondrial function and energy production and mitigates oxidative stress and inflammation, key contributors to disease pathology. The keto diet can reduce cerebral inflammation and the accumulation of Aβ and tau proteins, which are hallmark features of AD, potentially modifying disease progression and improving cognitive functions [210,211,212].

Recent research highlights the potential of physical exercise to modulate brain metabolism and improve cognitive function in AD. Aerobic exercise enhances mitochondrial function, reduces neuroinflammation, and decreases levels of Aβ and tau proteins in AD mouse models [213,214]. Exercise also increases the expression of neurotrophic factors like brain-derived neurotrophic factor (BDNF), which can improve cognition and alter amyloid precursor protein processing [215]. It improves glucose metabolism through pathways involving adenosine monophosphate-activated protein kinase (AMPK) and peroxisome proliferator-activated receptor gamma coactivator 1-alpha (PGC-1a) [186]. Studies in humans demonstrate that moderate-to-vigorous physical activity can improve learning, memory, and executive functions in healthy individuals and those with neurodegenerative disorders [216]. Multimodal exercise programs, combining aerobic, resistance, and balance training, show promise in improving activities of daily living in older adults with AD [217]. These benefits may be mediated through improved mitochondrial function, enhanced cerebral blood flow [218,219], components of the MNM hypothesis, and increased neuroplasticity [220]. These exercise-induced mechanisms suggest that lifestyle interventions could mitigate some of the metabolic impairments in AD, promoting neuroprotective pathways and improving cognitive outcomes [221].

### 6.5. Integration of Metabolic and Neurovascular Dysfunctions

The interconnectedness of cerebral vascular health and systemic metabolism is significant in understanding AD. Disruptions in NVC and metabolic pathways, such as impaired glucose metabolism and insulin resistance, are early indicators of AD. The mitochondrial dysfunction hypothesis ties into these aspects, highlighting how impaired mitochondrial function exacerbates metabolic and vascular abnormalities. Oxidative stress creates a detrimental feedback loop. This cycle contributes to a decline in mitochondrial function, further impairing cellular energy metabolism. Csiszar et al. conducted a study investigating treatments targeting the mitochondrial production of ROS [222]. Overexpression of human catalase in transgenic mice protects them from mitochondrial oxidative stress, supporting their hypothesis that antioxidative defenses provide cerebral microvascular protective effects [222]. Approaches that enhance mitochondrial efficiency and reduce oxidative stress show promise in addressing these dysfunctions. They offer a comprehensive strategy to slow AD progression, targeting metabolic and neurovascular pathways. Overall, the differences in brain metabolism between healthy individuals and those with AD highlight the critical role metabolic dysfunction plays in the disease’s pathogenesis. In the context of AD, early metabolic disturbances often precede noticeable cognitive decline. Impaired glucose metabolism, particularly in the hippocampus and cortex, reflects the initial metabolic breakdown that parallels vascular irregularities, such as decreased cerebral blood flow and BBB integrity. Reduced glucose uptake and utilization, insulin resistance, and mitochondrial dysfunction contribute to cognitive decline and neuronal damage. The implications of this research suggest that environmental factors may also play a role in AD, as well as steps that could be taken to avoid the disease’s development and progression into later clinical stages.

## 7. Therapeutic Strategies Addressing Components of the MNM Hypothesis

### 7.1. A Review of Current Therapeutic Approaches in Alzheimer’s Disease

While the direct path of AD progression remains unclear, neurodegeneration characterizes the disease, leading to progressive memory loss. No single pharmacological treatment prevents or cures AD. Overall, AD drug development has faced numerous challenges and failures over the past two decades, with a success rate of only 2% in phase II and III trials [223]. The amyloid hypothesis, which has dominated AD research, has been questioned due to repeated failures of amyloid-targeted therapies [59,63]. These failures have been attributed to various factors, including insufficient evidence for initiating pivotal trials, flawed trial designs, and substantial pre-symptomatic neuronal damage [32,223]. Several therapies, however, are available to mitigate disease progression and alleviate symptoms [224]. An overview of the current therapeutic strategies for AD, their mechanisms of action, targeted pathways, and limitations is provided in Table 2. In this section, we discuss current first-line treatment options for Alzheimer’s patients, their mechanisms of action (MOAs), and shortcomings to further elucidate the disease’s undetermined pathology.

Studies are exploring alternative approaches, such as tau-targeting therapies, vascular and mitochondrial dysfunction, neuroinflammation, and lifestyle factors [59,225]. Novel therapeutic strategies, including focused ultrasound, deep brain stimulation, and gene therapy, are being investigated alongside pharmaceuticals [32]. Developing effective disease-modifying treatments for AD remains an urgent need [226,227].

### 7.2. Amyloid-Targeting Therapies

The role of Aβ plaques in the pathogenesis of AD has yet to be fully elucidated, but their association with the disease is well established. As one of the prevailing explanations for the development of AD, the degradation of Aβ plaques has recently become another avenue of targeted therapy. Therapies such as Bapineuzumab and Aducanumab are monoclonal antibodies that work to reduce Aβ plaque loads. These drugs often show a reduction in Aβ plaques, but no significant changes to cognitive functions have been observed [228]. Furthermore, Aβ-targeting therapies for AD have largely failed to show clinical benefits in trials involving patients with mild to moderate AD [229]. Studies suggest that the timing of intervention is crucial, as anti-Aβ therapies may be more effective in delaying AD onset rather than treating symptomatic patients [230].

While anti-Aβ therapies have been extensively researched, their limited success in slowing cognitive decline has highlighted the need to explore alternative approaches. One potential way to overcome this limitation is through combination therapies that target multiple pathways implicated in AD. Since AD is a multifactorial disease, combining Aβ-targeting treatments with therapies addressing tau aggregation, neuroinflammation, and oxidative stress may yield more comprehensive results. Additionally, improved drug delivery techniques, such as nanotechnology, can enhance the penetration of Aβ-targeting agents across the BBB. Various nanoparticle types, including lipid-based, polymeric, and inorganic, have shown potential for enhancing drug transport to the brain [231,232]. Strategies such as surface modifications and specific targeting ligands improve drug delivery efficiency [233]. Timing is also critical: evidence suggests that early intervention, before significant neuronal damage has occurred, can improve cognitive outcomes, making early detection and preclinical treatment a priority [234]. Combining pharmacological treatments with lifestyle interventions like exercise and diet modification may also improve outcomes by addressing the metabolic dysregulation associated with AD [61].

To address the challenge of targeting multiple pathways, we propose a diagnostic sequence incorporating biomarkers and imaging techniques to guide personalized treatment strategies. Cerebrospinal fluid analysis of Aβ42, total tau, and phosphorylated tau can indicate amyloid deposition and tau pathology early on [235]. Following this, neuroimaging techniques such as amyloid PET scans offer a non-invasive way to visualize Aβ plaque deposition and tau neurofibrillary tangles. MRI can be utilized to assess hippocampal atrophy, which is a standard indicator of neuronal loss in AD [236]. Additionally, testing for neuroinflammatory markers like C-reactive protein and interleukin-6 could help determine the role of neuroinflammation in disease progression [237]. Assessing glucose metabolism through FDG-PET imaging and evaluating cerebral blood flow using dynamic susceptibility contrast MRI (DSC-MRI) could further highlight metabolic and vascular disruptions, critical components of AD pathophysiology [238]. Early genetic testing, such as screening for APOE ε4 status, could also be valuable in stratifying patient risk and guiding early interventions [239]. Timing is critical: evidence suggests that early intervention, before significant neuronal damage has occurred, can improve cognitive outcomes, making early detection and preclinical treatment a priority [234].

### 7.3. Cholinesterase Inhibitors and Cholinergic Agonists

Patients with AD have significant losses in choline acetyltransferase activity, leading to a marked decrease in acetylcholine levels and, subsequently, impaired cholinergic functioning [240,241]. One popular treatment option for patients is using cholinesterase inhibitors (ChEIs) such as donepezil, rivastigmine, and galantamine. These drugs bind to cholinesterase, the enzyme responsible for the breakdown of acetylcholine, inhibiting its action and subsequently allowing for a rise in acetylcholine [242]. ChEIs improve psychotic and cognitive symptoms in patients with AD [243,244], albeit modestly, and the literature does support a mortality benefit for patients on ChEIs long term [245]. When considering the MOA, these drugs enhance the availability of acetylcholine at the synapses, thereby facilitating neurotransmission within cholinergic pathways. The presence of therapeutic effects suggests that decreased levels of acetylcholine contribute to AD progression, as was first proposed in 1976 [246]; however, this hypothesis falls short in that treatment with ChEIs confers only minimal improvements in neuropsychiatric symptoms, suggesting further interplay between cholinergic and other processes.

Cholinergic agonists, such as varenicline and nicotine, are other processes that elicit cholinergic responses. Varenicline is a partial nicotinic cholinergic agonist that is traditionally used as an intervention for smoking cessation, reducing cravings and withdrawal symptoms [247]. Few clinical trials are aimed at utilizing varenicline’s MOA to improve cognitive function in those with AD and other related neurodegenerative diseases, with one having only a minor improvement compared to a placebo [223]. Further research is necessary to confirm the efficacy of varenicline in treating neurodegenerative diseases. The role of nicotinic acetylcholine receptors in central nervous system functioning has been widely described, and insufficient receptors likely contribute to the neural deterioration seen in many disease processes [248,249,250,251,252]. Nicotine stimulation of these receptors has, therefore, become a prevailing avenue of deriving therapeutic relief for those with neurodegenerative diseases. One such study explored using nicotine to reduce cognitive decline in those with Down syndrome [253]. Another study investigated the use of nicotine to improve cognitive impairment, but results are yet to be seen [254]. Data on the use of nicotine as it relates to facilitating brain functions varies, and further statistical analysis is needed before definitive results can be derived. The process of stimulating cholinergic receptors is not yet a definitive treatment for AD, suggesting that decreased acetylcholine and subsequent lack of receptor activation do not fully account for the pathogenesis of AD. Further work is warranted to elucidate the mechanisms underlying impaired neurocognition and the therapeutic effects of ChEIs and cholinergic agonists to identify novel corrective targets that may offer more comprehensive treatment strategies for neurodegenerative disorders.

### 7.4. Mitochondrial Dysfunction and Therapeutic Implications

The involvement of mitochondrial dysfunction in disease progression may explain the limited effectiveness of ChEIs in producing significant results. Mitochondria play a crucial role in cellular energy metabolism, including the production of acetyl-CoA, a precursor necessary for acetylcholine synthesis [255,256]. If mitochondrial dysfunction reduces acetyl-CoA production, ChEIs address only the downstream effects, limiting their ability to enhance cholinergic neurotransmission and leaving the underlying metabolic dysfunction unresolved. Current studies fail to show a rise in acetylcholine levels of those with AD on ChEIs compared to healthy individuals, and this may explain the limited therapeutic effects. Despite the limitations of ChEIs in addressing the broader spectrum of AD symptoms, they remain among the few pharmacological options with FDA approval for managing cognitive decline in patients.

### 7.5. NMDAR Antagonists and Calcium Dysregulation

An alternative to ChEIs and cholinergic agonists is the FDA-approved Alzheimer’s disease treatment memantine, an NMDAR antagonist. Glutamate activates NMDARs, gated ion channels that play a role in normal brain function [257]. The glutamatergic Alzheimer’s hypothesis suggests that in patients with Alzheimer’s disease, glutamate overstimulates the glutamatergic channels, leading to hyperactive NMDARs and neurotoxicity shortly after [258,259,260,261]. Inhibiting NMDARs quells unwarranted activity and protects against Ca^2+^-induced cell death [262]. Clinically, NMDAR antagonists do show promising results with improvements to cognition and functioning in patients with moderate to severe AD [263].

The effects of NMDAR antagonist treatment on those with mild AD showed no significant improvement compared to the placebo [263]. In mice studies, NMDAR antagonists showed marked improvements in various factors, including locomotion, spatial memory, Ca^2+^-dependent protein activation, and Aβ burden, validating memantine’s neuroprotective effects [264]. While NMDAR antagonists help suppress neurodegeneration, treatment alone does not stifle the progression of AD, indicating further interplay between pathways of disease development.

We suggest that rises in intracellular Ca^2+^ levels not only lead to cell death but also play a role in mitochondrial dysfunction, leading to the adversities of AD. Excessive Ca^2+^ influx can severely impair mitochondrial function [265,266]. Mitochondrial irregularities then contribute to disease progression through the down-synthesis of crucial neurotransmitters. Current oncological treatment studies hope to utilize Ca^2+^ overload to target mitochondrially active cancer cells as a therapy [267]. If successful, then the mechanism of NMDAR overactivation leading to increased intracellular Ca^2+^, thereby contributing to mitochondrial dysfunction, could explain the decreased incidence of cancer in patients with AD [268]. This relationship demonstrates the importance of targeting both synaptic and mitochondrial pathways in developing comprehensive treatment strategies for AD. It also suggests the likelihood that the pathogenesis and clinical presentation of AD are the result of the various cellular processes, including mitochondrial dysfunction. Despite focusing on neurotransmitter systems, targeting Aβ plaques remains a significant therapeutic avenue.

### 7.6. Integrating Multi-Targeted Approaches

Tolerance of drugs targeting Aβ plaques varies, but one commonality is their lack of efficacy. Some of the downfalls of these drugs can be attributed to the failure to breach the BBB. Focused ultrasound opens the BBB, enhancing effective drug delivery [269]. This technique has gained prominence in enabling more targeted treatments for various neurological diseases, making its application in treating AD a significant area of research. One such study aimed to combine focused ultrasound with the administration of aducanumab to clear Aβ in three patients. Focused ultrasound combination therapy has improved the specific removal of Aβ plaques, with significant reductions in areas of the brain exposed to opening and no change in unexposed areas [270]. While the clearance of Aβ was successful, the patients still experienced no improvements in cognition or behavior. This discrepancy suggests that Aβ accumulation may not be the primary driver of the disease’s progression, and it implies that Aβ plaques might be a downstream effect of other underlying pathological processes. The lack of symptomatic improvement following Aβ clearance challenges the amyloid hypothesis and indicates the necessity of exploring alternative mechanisms and therapeutic targets in AD. Recent studies further support this view, suggesting that Aβ may be a downstream effect of other processes, such as tau aggregation, neuroinflammation, and oxidative stress [147,271]. Although some anti-Aβ antibodies have shown promise in slowing disease progression [272], the inconsistency in cognitive improvements following Aβ clearance has led researchers to explore alternative pathways. Emerging studies propose that novel mechanisms, such as the T14 peptide, might act as upstream drivers of neurodegeneration [234], underscoring the complexity of AD pathophysiology and the need for a multifaceted treatment approach [167].

## 8. Integration of the MNM Components

While the amyloid cascade hypothesis has long dominated AD research, its limitations have become evident, particularly given the conflicting results of anti-Aβ therapies in clinical trials [273]. These shortcomings have prompted the exploration of alternative pathways, such as mitochondrial dysfunction, astrocyte activity, and neurovascular regulation. The MNM hypothesis builds on this shift and integrates these elements into a cohesive model that accounts for the multifactorial nature of AD. Unlike the amyloid-centric approach, the MNM hypothesis recognizes that mitochondrial impairment, neurovascular breakdown, and metabolic disturbances collectively drive AD progression, offering a more all-inclusive perspective on disease mechanisms. This approach aligns with recent research findings and provides a more precise direction for developing targeted, multi-faceted treatment strategies.

Based on the MNM hypothesis, exploring pathways contributing to AD pathology would be beneficial. One area includes the development of therapeutic strategies that target mitochondrial function. Considering the role of mitochondria in maintaining calcium homeostasis and contributing to neuronal energy metabolism, interventions aimed at improving mitochondrial dynamics, such as promoting fusion, inhibiting excessive fission, and reducing oxidative stress, may assist in preventing neurodegeneration. Therapeutic approaches that restore neurovascular health should also be investigated. This includes investigations into enhancing BBB integrity, improving CBF, and promoting angiogenesis via modifying endothelial cell function and astrocyte–neuron signaling. Furthermore, addressing systemic metabolic disturbances could also help understand the environmental factors contributing to AD. Investigating the role of insulin resistance in AD progression and evaluating the effectiveness of metabolic therapies, including the keto diet, could offer new possibilities for treatment. The potential for therapies that simultaneously target mitochondrial dysfunction, neurovascular regulation, and metabolic disturbances should be explored, considering the limitations of current therapeutic strategies, such as monoclonal antibodies targeting Aβ.

## 9. Future Directions and Validation of the Mitochondrial–Neurovascular–Metabolic Hypothesis

Current evidence from preclinical studies and small-scale clinical trials provides promising support for the MNM hypothesis [155]. Mitochondrial dysfunction, recognized as a central factor, contributes to oxidative stress and impaired bioenergetics that drive neuronal damage. Neurovascular impairment, including BBB disruption and reduced cerebral blood flow, similarly exacerbates cognitive decline, while glucose hypometabolism further compounds mitochondrial and neurovascular dysfunction.

Despite these findings, several critical areas warrant further empirical validation. First, while mitochondrial dysfunction’s role in neurodegeneration is well-supported, the precise mechanisms linking it to neurovascular impairment and metabolic dysregulation should involve detailed exploration. Notably, most existing research relies on animal models, which may not fully reflect human AD pathology [274]. Expanding human clinical trials that assess mitochondrial and neurovascular biomarkers, such as cerebral perfusion and BBB integrity, is therefore necessary to establish relevance in human disease. Further, as emerging evidence suggests, sex differences may influence mitochondrial and neurovascular pathways, calling for future studies to examine these variations to ensure that therapeutic approaches are tailored appropriately [275]. Lastly, while the MNM hypothesis shows promise for early-stage AD, its relevance in more advanced stages remains to be validated. Larger-scale and longitudinal clinical studies following AD patients through various disease stages will clarify whether mitochondrial and neurovascular interventions can yield sustained benefits across the disease continuum [276,277]. Strengthening the clinical and mechanistic foundation of the MNM hypothesis could guide the development of multi-targeted interventions that address AD’s complex pathology and potentially improve patient outcomes at various disease stages.

## 10. Conclusions

The MNM hypothesis offers a transformative and comprehensive overview that intricately integrates mitochondrial dysfunction, neurovascular dysregulation, and impaired glucose metabolism to explain the pathogenesis of AD. Although several hypotheses have been developed to describe the development and progression of this debilitating disease, the MNM hypothesis uniquely combines theories into a cohesive explanation. Our review stresses the critical role of altered brain metabolism in causing a cascade of pathological events leading to AD. This comprehensive framework addresses the Ineffectiveness of single-target therapies, which predominately focus on reducing the Aβ accumulation with limited success.

Our perspective highlights how increased insulin resistance and reduced glucose uptake alter brain metabolism and initiate a cascade of pathological events that exacerbate mitochondrial impairments and ROS formation. This metabolic dysfunction disrupts neurovascular integrity, eventually resulting in BBB breakdown and neurotoxic infiltration into the brain. The interconnectedness of these processes, often considered separate, highlights the urgent need to move beyond the traditional amyloid hypothesis to develop new interventions targeting the complex interplay of metabolic, mitochondrial, and vascular dysfunctions.

Future research directions should focus on developing therapeutic strategies that enhance mitochondrial function, reduce oxidative stress, and restore neurovascular integrity. Combining these approaches calls for the development of more holistic treatments for AD. Investigations into lifestyle interventions such as exercise and dietary modifications like the ketogenic diet also show potential in modifying the metabolic pathways implicated in AD and provide novel avenues in treatment. Examining the role of biomarkers for mitochondrial and vascular health in combination with neuroimaging can also lead to early detection methods and personalized intervention strategies. The MNM hypothesis opens new opportunities for comprehensive and multifactorial interventions, directing the field toward therapies more accurately addressing the disease’s underlying mechanisms, and offering a stronger foundation for preventive and therapeutic strategies.

In conclusion, the MNM hypothesis provides a multifaceted understanding of AD pathogenesis, emphasizing the relations between metabolic, mitochondrial, and neurovascular dysfunctions. Addressing these interconnected pathways allows future research and therapeutic approaches to target the root causes of AD more effectively, leading to better prevention and treatment strategies. This comprehensive understanding paves the way for groundbreaking interventions that could significantly alter the trajectory of AD, improving outcomes for millions of affected individuals.

## Figures and Tables

**Figure 1 ijms-25-11720-f001:**
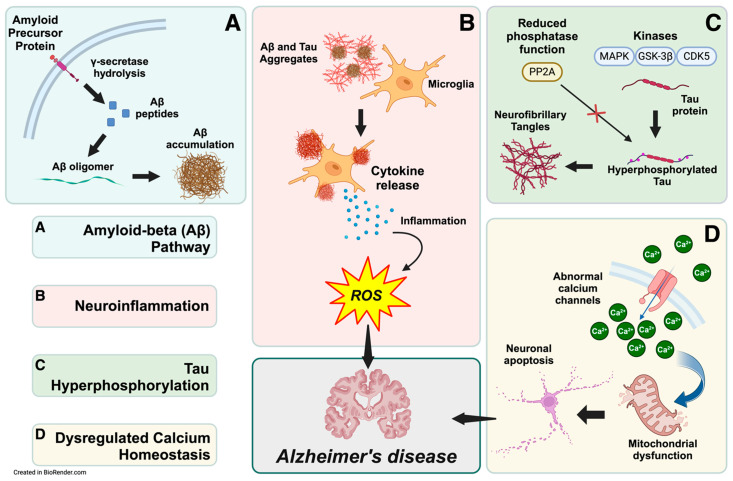
Biochemical pathways leading to Alzheimer’s disease pathogenesis. The interconnected biochemical pathways implicated in Alzheimer’s disease (AD) development include the amyloid-beta (Aβ) pathway, neuroinflammation, tau hyperphosphorylation, calcium homeostasis dysregulation, and mitochondrial dysfunction. (**A**) In the Aβ pathway, γ-secretase cleaves amyloid precursor protein within neurons to produce Aβ peptides. These peptides are subsequently released into the extracellular space, accumulating and aggregating into Aβ oligomers and plaques. (**B**) The extracellular Aβ aggregates activate microglia, releasing pro-inflammatory cytokines and reactive oxygen species (ROS), which sustain neuroinflammation and oxidative stress. (**C**) In parallel, kinases like CDK5, GSK-3β, and MAPK hyperphosphorylate the tau protein. Reduced phosphatase activity also contributes to tau hyperphosphorylation and forms intracellular neurofibrillary tangles, contributing to neuronal dysfunction. (**D**) Dysregulation of calcium channels disrupts calcium homeostasis, increasing intracellular calcium levels and impairing mitochondrial function, further promoting neuronal apoptosis. Together, these pathways converge, leading to neurodegeneration and cognitive decline characteristic of AD. This integrated view highlights the multifactorial etiology of AD and the therapeutic potential of targeting these interrelated mechanisms.

**Figure 2 ijms-25-11720-f002:**
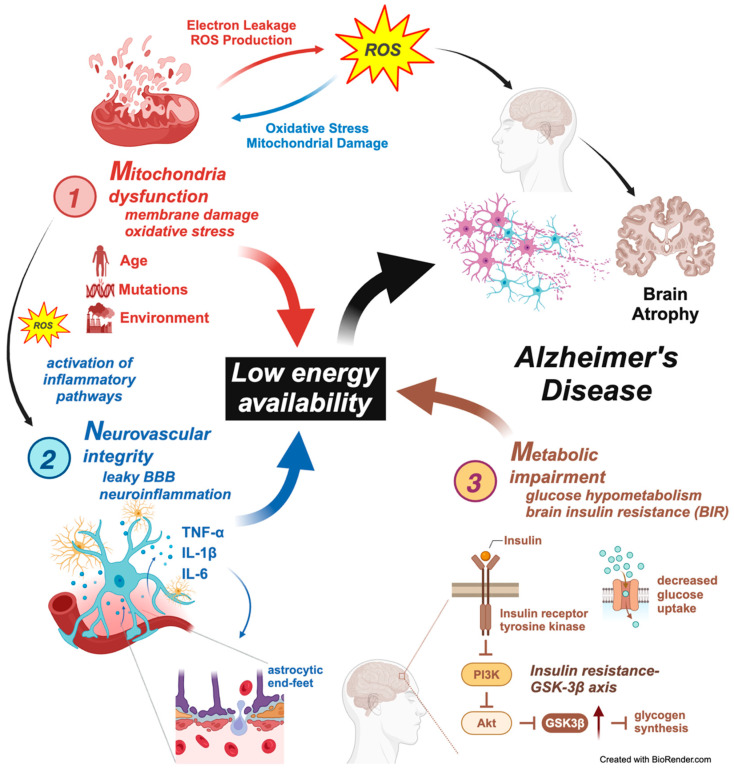
The Mitochondrial–Neurovascular–Metabolic (MNM) hypothesis of Alzheimer’s disease pathogenesis. This figure illustrates the interconnected pathways of the Mitochondrial–Neurovascular–Metabolic (MNM) hypothesis, which proposes a multifaceted approach to understanding Alzheimer’s disease (AD) pathogenesis. The model emphasizes how mitochondrial dysfunction, neurovascular integrity disruption, and metabolic impairment contribute to a state of low energy availability, ultimately leading to AD progression. Three components are involved as follows. 1. Mitochondrial dysfunction—mitochondria damage and oxidative stress: Mitochondrial dysfunction increases the production of reactive oxygen species (ROS), leading to oxidative stress and membrane damage. Electron leakage and excessive ROS generation contribute to mitochondrial damage, promoting a cycle of oxidative injury and impaired mitochondrial function. Activation of inflammatory pathways: As mitochondrial dysfunction progresses, the release of ROS activates inflammatory pathways, promoting neuroinflammation. Factors such as aging, genetic mutations, and environmental influences exacerbate mitochondrial damage. The resulting oxidative stress further contributes to neuronal damage and the overall decline in brain function seen in AD. 2. Neurovascular integrity disruption—leaky BBB and neuroinflammation: The neurovascular unit, comprising endothelial cells, pericytes, astrocytes, and neurons, plays a crucial role in maintaining blood–brain barrier (BBB) integrity. In AD, activating inflammatory cytokines such as TNF-α, IL-1β, and IL-6 leads to disruption of the BBB, allowing harmful substances to enter the brain. Astrocytic end-feet disruption: Damage to the astrocytic end-feet surrounding blood vessels contributes to BBB breakdown, further exacerbating neuroinflammation. This process perpetuates a cycle of neuronal injury, oxidative stress, and energy deficits. Compromised BBB integrity increases neuroinflammation, accelerating mitochondrial damage and contributing to AD pathogenesis. 3. Metabolic impairment—glucose hypometabolism and brain insulin resistance (BIR): The figure shows that metabolic impairments, including glucose hypometabolism and insulin resistance, are central to AD progression. Insulin signaling disruption leads to reduced PI3K/Akt pathway activation, which is critical in neuronal survival, energy metabolism, and synaptic plasticity. Insulin resistance–GSK-3β axis: Insulin resistance results in reduced glucose uptake and impaired activation of the PI3K/Akt pathway, leading to increased GSK-3β activity. This enzyme contributes to tau hyperphosphorylation and glycogen synthesis, worsening the energy deficit and contributing to AD progression. The reduced availability of glucose for neuronal energy production further drives neurodegeneration and cognitive decline in AD. Central role of low energy availability: The culmination of mitochondrial dysfunction, neurovascular disruption, and metabolic impairment leads to low energy availability, which is central to the MNM hypothesis. This energy deficit triggers widespread neuronal damage, synaptic dysfunction, and ultimately, brain atrophy, characteristic of Alzheimer’s disease. Integration with Alzheimer’s disease pathogenesis: The figure highlights how the convergence of mitochondrial, neurovascular, and metabolic disturbances culminates in AD progression, leading to characteristic features such as brain atrophy, amyloid-beta accumulation, and tau pathology. This figure comprehensively captures the MNM hypothesis’s interconnected pathways, demonstrating how mitochondrial dysfunction, neurovascular integrity disruption, and metabolic impairment collectively drive the pathogenesis of Alzheimer’s disease. The model underscores the need for multifaceted therapeutic approaches that target these interrelated mechanisms to combat AD progression effectively.

**Figure 3 ijms-25-11720-f003:**
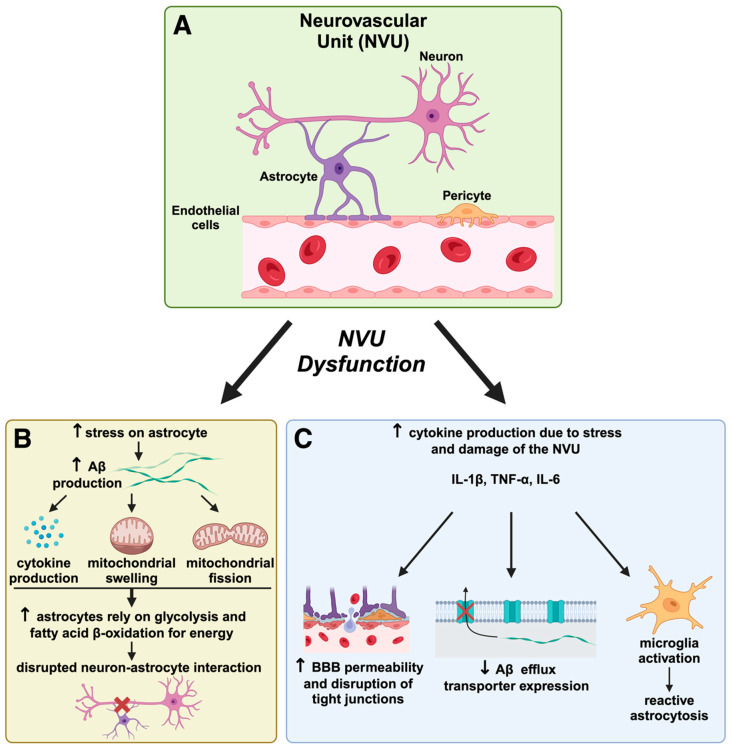
Neurovascular unit dysfunction and its role in Alzheimer’s disease pathogenesis. The neurovascular unit (NVU) is a complex network comprising neurons, astrocytes, endothelial cells, and pericytes that maintains blood–brain barrier (BBB) integrity and regulates cerebral homeostasis. In AD, NVU dysfunction leads to a cascade of harmful effects that exacerbate disease progression. (**A**) The neurovascular unit (NVU) and initial disruption. The NVU, shown here in a healthy state, consists of neurons, astrocytes, pericytes, and endothelial cells that work together to support BBB integrity and regulate cerebral blood flow. In AD, NVU integrity is compromised, initiating processes that lead to neuronal damage. NVU dysfunction contributes to BBB breakdown, allowing potentially neurotoxic substances to enter the brain, which accelerates neuroinflammation and oxidative stress, as suggested by the Mitochondrial–Neurovascular–Metabolic (MNM) hypothesis discussed in our review. (**B**) Astrocyte dysfunction and Aβ accumulation. Astrocytes under stress produce cytokines and undergo mitochondrial swelling and fission, leading to increased amyloid-beta (Aβ) production. This results in altered astrocyte metabolism, where astrocytes shift to glycolysis and fatty acid β-oxidation for energy production due to mitochondrial impairment. The figure highlights how this metabolic shift disrupts the critical neuron–astrocyte interaction, depriving neurons of essential support and contributing to AD pathology by furthering Aβ aggregation and neurotoxicity. This aligns with the MNM hypothesis, which posits that metabolic disturbances play a central role in AD progression by impairing cellular energetics. (**C**) Cytokine production, BBB permeability, and reactive astrogliosis. NVU damage increases inflammatory cytokine production (e.g., IL-1β, TNF-α, IL-6), further weakening the BBB by disrupting tight junctions. This facilitates greater BBB permeability and impairs the efflux of Aβ, resulting in its accumulation in the brain parenchyma. Reactive astrogliosis and microglial activation are inflammatory responses, creating a feedback loop that exacerbates neuronal damage. This progression supports the MNM hypothesis by demonstrating how neurovascular breakdown and inflammation intertwine metabolic and mitochondrial dysfunctions in AD. This figure illustrates critical concepts of the MNM hypothesis proposed in our manuscript, which argues that AD results from interconnected pathways involving mitochondrial dysfunction, neurovascular dysregulation, and metabolic impairment. The NVU dysfunction depicted here exemplifies how neurovascular compromise can accelerate AD by promoting Aβ accumulation, oxidative stress, and inflammation. Therapeutic strategies targeting NVU integrity, inflammation, and metabolic health could be promising avenues for disease modification in AD.

**Figure 4 ijms-25-11720-f004:**
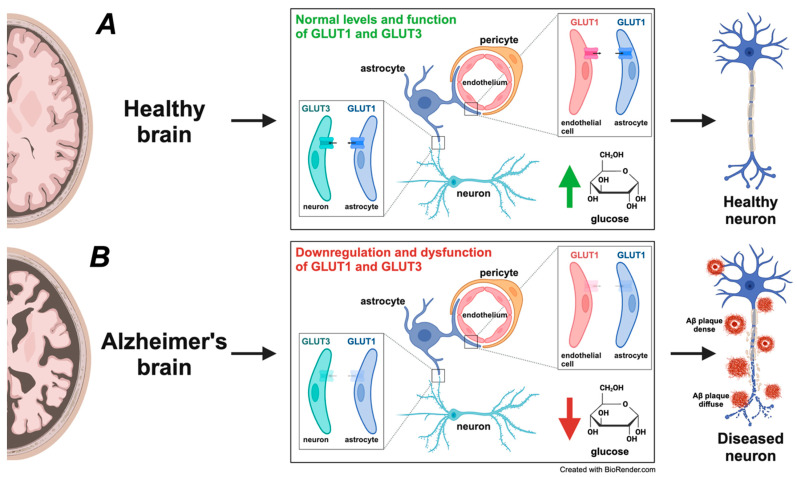
Comparative analysis of glucose transporter expression and glucose metabolism in healthy versus Alzheimer’s disease (AD) brains. (**A**) In a healthy brain, glucose is the primary energy source, and efficient glucose metabolism is critical for maintaining normal neuronal function and cognitive processes. Glucose transporters GLUT1 and GLUT3 play a central role in facilitating glucose uptake. GLUT1, primarily expressed on endothelial cells of the blood–brain barrier (BBB) and astrocytes, ensures a steady influx of glucose into the brain’s parenchyma. GLUT3, present in neurons, provides glucose for neuronal energy demands, supporting synaptic activity, neurotransmitter synthesis, and overall neuronal health. This efficient glucose uptake and metabolism sustain neurons’ healthy structure and function, depicted as the healthy neuron on the right. (**B**) In the AD brain, a notable downregulation and dysfunction of glucose transporters GLUT1 and GLUT3 occurs, depicted by their transparent appearance, indicating reduced expression and impaired function. This downregulation and dysfunction decrease glucose uptake and hypometabolism, leading to compromised energy production, increased neuronal stress, and synaptic dysfunction. The impaired glucose transport causes an energy deficit, ultimately contributing to amyloid-beta (Aβ) plaque formation and tau pathology, hallmark AD features surrounding the diseased neuron (diffuse and dense Aβ plaques). The metabolic disturbance accelerates neurodegeneration and the progression of AD. This figure underscores the pivotal role of glucose metabolism in AD pathogenesis, underscoring how the diminished expression and malfunction of GLUT1 and GLUT3 contribute to neuronal dysfunction, synaptic impairment, and overall neurodegeneration. These alterations align with the Mitochondrial–Neurovascular–Metabolic (MNM) hypothesis discussed in the paper, demonstrating how disruptions in metabolic pathways play a central role in AD development. The significance of these findings provides crucial insights into the mechanisms underlying AD.

**Table 1 ijms-25-11720-t001:** Comparison of amyloid-beta and MNM hypotheses in the pathology of Alzheimer’s disease.

Feature	Amyloid-Beta (Aβ) Hypothesis	Mitochondrial–Neurovascular–Metabolic (MNM) Hypothesis	Detection Methods
**Main Postulate**	Accumulation of amyloid-β (Aβ) peptides triggers AD pathogenesis	AD pathogenesis involves mitochondrial dysfunction, neurovascular dysregulation, and metabolic disturbances	**Aβ hypothesis**: immunohistochemistry, PET scans with PiB
			**MNM hypothesis**: mass spectrometry, magnetic resonance spectroscopy (MRS)
**Pathogenesis**	Aβ accumulation leads to neurodegeneration and cognitive decline	Combined impact of mitochondrial damage, vascular issues, and metabolic imbalance on neural health	**Aβ hypothesis**: Western blot for Aβ oligomers, ELISA
			**MNM hypothesis**: oxygen consumption rates, ATP assays
**Key Evidence**	-Presence of Aβ plaques in AD brains	-Mitochondrial dysfunction observed in AD patients	**Aβ hypothesis**: PET/MRI for Aβ plaques
	-Genetic mutations in *APP*, *PSEN1*, and *PSEN2* linked to early-onset AD	-Impaired BBB integrity	**MNM hypothesis**: oxygen consumption rates, ATP production assays
		-Metabolic disturbances such as glucose hypometabolism	
**Therapeutic Implications**	-Aβ-targeting drugs (e.g., monoclonal antibodies)	-Potential therapies targeting mitochondrial health, neurovascular integrity, and metabolic regulation	N/A
	-Limited efficacy and clinical benefits	-Comprehensive, multi-target approaches	
**Limitations**	-Inconsistent therapeutic success with Aβ-targeting drugs	-Complexity in targeting multiple pathways simultaneously	N/A
	-Does not account for all AD etiologies, including genetic, environmental, and metabolic factors	-Requires further empirical validation	
		-Interactions between pathways not yet fully understood	
**Strengths**	-Extensive research and numerous clinical trials	-Integrative approach considering multiple pathways	N/A
	-Some approved drugs (e.g., aducanumab)	-Addresses limitations of single-target strategies	
**Weaknesses**	-Limited success in clinical trials	-More complex, requiring multifaceted interventions	N/A
	-Focuses mainly on Aβ, potentially overlooking other factors	-Less established research compared to Aβ hypothesis	

**Table 2 ijms-25-11720-t002:** Therapeutic strategies for Alzheimer’s disease.

Therapeutic Strategy	Mechanism of Action	Targeted Pathways	Limitations
Monoclonal Antibodies (e.g., Aducanumab)	Target amyloid-β plaques and facilitate their clearance	Amyloid-β pathway	Limited cognitive improvements and high cost
Cholinesterase Inhibitors (ChEIs)	Inhibit breakdown of acetylcholine, enhancing cholinergic neurotransmission	Cholinergic system	Modest efficacy with minimal alterations to disease course
Cholinergic agonists	Stimulate cholinergic receptors	Cholinergic system	Not yet approved as a treatment for AD, limited data on efficacy
NMDA receptor antagonists	Reduce glutamate-induced excitotoxicity	Glutamatergic system	Effective in moderate to severe AD, limited benefit in mild AD
Metabolic Therapies (e.g., Ketogenic Diet)	Shift brain energy source from glucose to ketone bodies	Metabolic regulation	Requires strict dietary adherence, long-term effects unclear
Mitochondrial-targeted Therapies	Enhance mitochondrial function, reduce oxidative stress	Mitochondrial health	Experimental, need for further validation
Lifestyle Interventions (e.g., Exercise)	Improve mitochondrial function, increase neurotrophic factors	Metabolic and neurovascular pathways	Requires consistent patient compliance, variable outcomes

## Data Availability

No new data were created or analyzed in this study. Data sharing is not applicable to this article.

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
