# Peer review of "From Plaques to Pathways in Alzheimer’s Disease: The Mitochondrial-Neurovascular-Metabolic Hypothesis"

_ijms, 2024, doi:10.3390/ijms252111720_

Round 1

Reviewer 1 Report

Comments and Suggestions for Authors

The manuscript titled “From Plaques to Pathways in Alzheimer’s Disease: Integrating Mitochondrial, Neurovascular, and Metabolic Perspectives” by Kazemeini, S.; et al. is a Review work where the authors outlined the most recent advances in pivotal action of the mitochondrial, metabolic and neurovascular crosstalk and how they impact on the onset and progression of Alzheimer’s disease. This is a topic of growing importance and the manuscript is generally well-written. However, it exists some points that need to be addressed (please, see them below detailed point-by-point) to improve the scientific quality of the submitted manuscript paper before this article will be consider for its publication in the International Journal of Molecular Sciences.

1) “Alzheimer’s Disease (…) trillion dollars in 2050” (page 1). It would be also beneficial to provide some data insights according to the life expectancy according the sex and race (in order to prevent any bias). This will greatly aid the potential readers to better understand the significance of this Review work.

2) “The understanding of AD (…) identifying neurofibrillary tangles and plaques (…) pathology involves amyloid-beta (…) tau hyperphosphorylation, and mitochondrial dysfunction (…) progression” (pages 1-2). Here, even if I agree with these statements provided by the authors it should be not neglected the contribution of other fibrils formed by the S100A family [1] or alpha-synuclein [2] which are strongly influenced by the ionic concentration and redox conditions inner the brain tissues.

[1] Carapeto, A.P.; et al. Morphological and Biophysical Study of S100A9 Protein Fibrils by Atomic Force Microscopy Imaging and Nanomechanical Analysis. Biomolecules 2024, 14, 1091. https://doi.org/10.3390/biom14091091

[2] Ziaunys, M.; et al. Polymorphism of Alpha-Synuclein Amyloid Fibrils Depends on Ionic Strength and Protein Concentration. Int. J. Mol. Sci. 2021, 22, 12382. https://doi.org/10.3390/ijms222212382

3) “2.1. Biochemical mechanisms and pathways” (page 2). A schematic representation should be depicted in order to better visualize all the existing biochemical routes that could lead to the Alzheimer’s disease. Similar comment for the section “5. Neurovascular Dysregulation in Alzheimer’s Disease” (pages 9-12).

4) Table 1 (page 5). A extra column should be added to discuss about the potential techniques to detect the presence of amyloid-beta fibrils and plaques and mitochondrial metabolites.

5) “Conclusion” (pages 19-20). This section perfectly remarks the most relevant outcomes found by the authors in this field and also the promising future perspectives. It should be also mentioned the potential future action lines to pursue the topic covered in this research.

Reviewer 2 Report

Comments and Suggestions for Authors

This paper by Vida and colleagues focused on the overview of Alzheimer’s disease which is associated with Memory loss and cognitive decline, a progressive neurological illness. Despite its dominance, the amyloid cascade hypothesis—which centers on amyloid-beta (Aβ) peptides—has had only patchy success in clinical studies. This paper focused on Mitochondrial-Neurovascular-Metabolic (MNM), which links metabolic abnormalities, neurovascular dysregulation, and mitochondrial dysfunction as interrelated factors in the pathophysiology of AD. This report also highlights about the mitochondrial Dysfunction and Neurovascular Dysregulation and highlights the need of treating and comprehending Alzheimer's disease from a variety of angles. The paragraphing is concise and simple to understand. I have a few suggestions and questions regarding this work. Addressing these questions could further enrich the study findings and provide a more comprehensive understanding of AD and ideal treatment strategies.

1.       Page 19 “While the clearance of Aβ was successful, the patients still experienced no improvements in cognition or behavior. This discrepancy suggests that Aβ accumulation may not be the primary driver of the disease’s progression, and it implies that Aβ plaques might be a downstream effect of other under lying pathological processes. Lack of symptomatic improvement following Aβ clearance challenges the amyloid hypothesis and indicates the necessity of exploring alternative mechanisms and therapeutic targets in AD.”

The authors need to cite relevant literature or evidence for this statement.

2.       Unpredictable Therapeutic Outcomes: Clinical investigations have demonstrated that anti-amyloid-beta (Aβ) treatments have limited efficacy, failing to appreciably reduce cognitive deterioration. Can the authors briefly describe how this limitation could be overcome?

3.       Complexity of Targeting Multiple Pathways: The complex etiology of Alzheimer’s Disease (AD) complicates the development of treatments that simultaneously target all contributing components. Do the authors propose a particular diagnostic sequence to tackle this challenge? If affirmative, please incorporate into the discussion section.

4.       Need for Further Validation: Requirement for Additional Validation: The Mitochondrial-Neurovascular-Metabolic (MNM) hypothesis necessitates more empirical confirmation to ascertain its efficacy. Can the authors elaborate this section and discuss in detail about the necessary steps.

Interactions Not Fully Understood: The intricate relationships among mitochondrial malfunction, neurovascular dysregulation, and metabolic impairment remain inadequately comprehended, hindering therapeutic advancement. The authors need to 

Reviewer 3 Report

Comments and Suggestions for Authors

A well contained manuscript. However, Diets and ethnicity in developing ALZ and their possible social determinants and their interactive pathways  is somehow overlooked, which should be included, along with the social statute.  Also, what is the specific contribution and the work in the ALZ field by the these contributors has to be highlighted. 
